# Architectural Organization of Dinoflagellate Liquid Crystalline Chromosomes

**DOI:** 10.3390/microorganisms7020027

**Published:** 2019-01-22

**Authors:** Joseph Tin Yum Wong

**Affiliations:** Division of Life Science, Hong Kong University of Life Science, Clearwater Bay, Kowloon, Hong Kong; botin@ust.hk; Tel: 852-23587343

**Keywords:** dinoflagellate, liquid crystalline chromosomes, liquid crystalline DNA, phase transition, superhelical module, 5-hydroxymethyluracil, nucleosome, superhelical condensation, chromosome

## Abstract

Dinoflagellates have some of the largest genome sizes, but lack architectural nucleosomes. Their liquid crystalline chromosomes (LCCs) are the only non-architectural protein-mediated chromosome packaging systems, having high degrees of DNA superhelicity, liquid crystalline condensation and high levels of chromosomal divalent cations. Recent observations on the reversible decompaction–recompaction of higher-order structures implicated that LCCs are composed of superhelical modules (SPMs) comprising highly supercoiled DNA. Orientated polarizing light photomicrography suggested the presence of three compartments with different packaging DNA density in LCCs. Recent and previous biophysical data suggest that LCCs are composed of: (a) the highly birefringent inner core compartment (i) with a high-density columnar-hexagonal mesophase (CH-m); (b) the lower-density core surface compartment (ii.1) consisting of a spiraling chromonema; (c) the birefringent-negative periphery compartment (ii.2) comprising peripheral chromosomal loops. C(ii.1) and C(ii.2) are in dynamic equilibrium, and can merge into a single compartment during dinomitosis, regulated through multiphasic reversible soft-matter phase transitions.

## 1. Introduction

Dinoflagellate liquid crystalline chromosomes (LCCs) are strongly birefringent, when observed under polarizing light microscopy (live cells without fixation) [1,2], suggesting a highly anisotropic organization of condensed DNA in vivo (see Figure 1a). LCCs have no architectural nucleosomes [3,4,5] and low concentrations of acid-soluble chromosomal proteins [6], despite having some of the largest genomes [7]. Instead of “beads-on-string” nucleosomal structures [8], relaxed periodic helical plectonemes were observed in dinoflagellate chromosome spread preparations [9,10], consistent with decondensed modular domains of superhelical DNA.

## 2. Superhelical Plectonemic Modules (SPMs) and Compartmentation

Cation-chelation (1-6mM EDTA) induced orchestrated concentration-dependent decompaction of LCCs, and the associated loss of higher order (at 1-2mM, lost of birefringence) could be partially repacked with the reintroduction of divalent cations [14]. This finding contrasted with the relatively milder effect of EDTA treatment on nc-chromosomes [15], which was likely related to the 2- to 3-fold increased LCC Mg^2+^ concentrations [16]. Mg^2+^ could mediate the condensation of supercoiled DNA, but on its own would not incur reversibility in the soft-matter phase transition [17], which would be required for chromosome operations [18]. Liquid crystalline phase transitions of supercoiled DNA were governed by physical-mechanical association between modules, which in turn were influenced by multitudes of physical parameters and subunit configurations [19,20,21,22,23,24]. Reversible decompaction implicated that the superhelical LCCs higher-order structures were composed of modular superhelical modules (SPMs) anisotropically organized, and was in agreement with a major role of inter-strand divalent cations in mediating lc-mesophases. “Rosette-like” ultrastructure structures [25], observed in two types of partially decompacted LCCs, were consistent with SPMs (Figure 2).

Transmission electron microscopy (TEM) and DNA fluorescent microscopy of chelator-treated LCCs (5-6mM EDTA), revealed a decompacted screw thread-like appearance surrounding an extended central core [14]. This despondent effect of C(i) relative to C(ii.1) was supportive of the two compartments having different compaction and major connections only at chromosome ends. It was also in agreement with the right-angle relationship between the helical moments in C(i) and C(ii.1), which was suggested by black and white birefringence (Figure 1a., suggesting opposite orientation), as well as the higher retardance in C(i) in relation to C(ii.1)) and C(ii.2) (Figure 1b–d).

Under physiological concentration ranges of DNA, the cholesteric mesophase (Cm) is the common liquid crystalline mesophase encountered, followed by columnar-hexagonal phase (C-Hm) with higher density than [26]. A cholesteric circular dichroism (CD) signal at -265 nm from outer condensed region of EtBr-tagged dinokaryon, but not in nascent CD spectra [27]. This signal is supportive of cholesteric phase to a certain organization level of c(ii.1), and C-Hm to c(i), which had a higher retardance. The EtBr-dependent signal was misinterpreted as a higher-order cholesteric organization without regarding the DNA-intercalation effect of EtBr. This lack of a 265-nm signal in the nascent CD spectrum, but only in the mildly intercalated EtBr spectrum suggested that the cholesteric signal likely corresponded to sub-chromonema level of c(ii.1) and not to a higher order on the LCC periphery. The high scattering CD values [27] confirmed the non-uniformity of the liquid crystalline phases (i.e., between c(i) and c(ii.1)), and not related to a protein core as originally interpreted.

Gene encoding region was the only fraction accessible for restriction enzyme digestion [28]. The birefringent negative C(ii.2), indicative of decondensed DNAs, corresponded to outer semi-condensed peripheral chromosome loops (PCLs) being the only LCC sub-location that was transcriptionally active[29,30]. Significantly, this implies global transcriptional regulation would be mediated through multiphasic soft-matter phase transitions of PCL, between isotropic and liquid crystalline phases of DNA. Mn^2+^ was a stronger liquid crystalline cations then Mg^2+^ [31]; Mn^2+^ stimulated transcription in other eukaryotic cells, inhibited transcription in dinokaryon [6]. PCL could possibly condense, especially during mitosis, and be considered part of C(ii.1).

Based on the above interpretation of data, the Compartmental Superhelical Liquid Crystalline Model (CSLCM) is proposed:

(a) LCCs are composed of modular units termed superhelical plectonemic modules (SPMs), which are formed by highly supercoiled DNA;

(b) LCCs compose of an inner core C(i), a core surface C(ii.1) in dynamical equilibrium with C(ii.2) comprising of transcriptionally-active periphery chromosomal loops (PCL).

The chromonema is composed of coil-spiralling SPMs in tandem (Figure 3). The despondent central compacted core surrounded by an external core-spiral, after EDTA- chelation ex vivo [14], was consistent with chromonema in C(i) and C(ii.1-ii.2) coiling independenly, with C(i) coiling on its own axis (with higher density) into C-Hm formation of SPMs and C(ii.1) chromonema coils around C(i), which would incur less density and in the front line of decompaction in response to chelation. The central C(i), with higher DNA-packaged density, thus acts as a structural carrier [29] for PCL, which would be in dynamical phase equilibrium with C(ii.1). 

LCC decompaction and remodeling were induced with a sample preparation of ultrastructural studies of dinokaryons and attributed to an insufficiency of DNA-protein chemical cross-links associated with a very low protein-DNA ratio [33]. The decondensation of supercoiled domains with this lack of fixation, together with physical severance at ultrasectioning, reasonably incurred the commonly observed “nested-arch” structures [34] with common TEM preparation protocols. Given that mathematical deduction from this “higher-order waveform” was the basis of the previously proposed cholesteric model [35], the accompanying interpretation of a single cholesterite could not be substantiated; Ethanol (30%) was the fixative agent during the sample preparation for SIMS imaging of the divalent cation [16], however, ethanol (>20%) induced different structures in non-nucleosomal DNAs [36]. The secondary ion mass spectroscopic-imaged holes were likely to be actual space, incurred by ethanol-induced DNA structural changes, and an architectural protein core would not be consistent with low basic chromosomal protein concentrations [37,38]. Ethanol exhibited a conservation effect on LCC Mg^2+^ concentrations, indicating that a proportion of higher divalent cation was attributed to counterions, which would not be disrupted by ethanol.

## 3. Physical Genomic Karyotype (PGK) of Liquid Crystalline Chromosomes

Genomes and associated molecular processes have to be optimally crafted for the physical realization of the genome carrier. There was a reduced diversity of specific transcription factors and DNA-binding proteins, but increase in predicted RNA-binding proteins, consistent with post-transcriptional regulation of gene expression (reviewed in [38]); genes were unidirectionally encoded and many were in tandem array genes [39,40,41]. Compared to the common state in nc-chromosomes, this set-up will facilitate the avoidance of large and sustained transcriptional bubbles, as well as dimensionally-equalizing transcriptional units, both resulting in increased anisotropy and a higher propensity for soft-matter phase transitions, and configurative periodicity. Conversely, transcriptional units with different dimensions and cross-transcriptional connections would be detrimental to anisotropic phase transitions. Transcription along unidirectional tandem arrays will be logistically conducted with immediate closure instead of keeping the transcription bubble “open” for sustained transcription over the same gene loci, minimizing both the duration and size of non-anisotropy. Putative long tandem-array transcripts will further ensure the avoidance of trans-splicing in large transcriptional bubbles at inconvenient times. These physical optimizations of the genome at all levels of chromosomal organization (including the non-random distribution of 5Hmu), especially in the configurative well-being of LCCs for reversible soft matter phase transitions, cumulated to the Physical Genomic Karyotype(PGK); a global reduction in modular configurative variability and an increase in three-dimensional modularity content with post-transcriptional regulation of gene expression through soft-matter phase transition. These are evidently one primary mission of PGK. A fully-annotated dinoflagellate chromosomal genome sequence would be revealing to this additional dimension of organization.

A special dinoflagellate RNA leader sequence (CSL) was identified [42,43]. The phase separation of RNA granules was influenced by RNA lengths, secondary structures, and concentrations [44,45,46], and could potentially be configured with potential CSL-binding proteins or spacer-binding proteins. Inter-ORF spacers were only rarely found in transcriptome [47], likely attributed to special compartmentation that eluded common RNA preparation method; such spacers would potentially confer periodicity on RNA secondary structures [48,49], potentially guiding transcription and splicing through soft matter phase transitions, and affecting the expression dynamics of corresponding transcripts. These arrangements are consistent with the general transcription “rate” not being dominantly regulated at the level of promotor elements, but controlled at pre-, and post-transcriptional gene topology level, critically through the phase transition of nucleic acid domains, and the post-transcriptional processing of transcripts. 

## 4. 5-Hydroxymethyluracil and Z-DNA

A substantial amount of dinoflagellate genomic thymine (60–70%) was replaced by 5-hydroxymethyluracil (5Hmu) [50,51,52], which exhibited a density-dependent distribution in the CsCl gradient [50], implicating possible distinctive compartmental distribution. As DNA supercoiling would be preserved in such a condition, it implicated 5Hmu enrichment in C(i), which had the higher-density C-Hm. The 5Hmu increased DNA flexibility through increased propensity for a roll and tilt between bases, especially when interspersed with G-C pairs [53,54,55]. The reduction of DNA structural water (_c_water) with high concentrations of divalent counterions, which commonly led to reduce DNA rigidity [56], would further increase the propensity for compartment-specific DNA supercoiling. dsDNA pair potential, 5Hmu placement, and high levels of counterions would be synergistic in raising DNA twistability, likely resulting in an increased propensity for supercoiling and SPM compaction. Cooperative supercoiling-mediated compaction of SPMs will increase the overall anisotropic configurability. A·T-rich dinucleotide steps increased DNA flexibility [56,57]; extrapolation of similar sequence-dependency to A·H-rich dinucleotide steps would further increase DNA flexibility. Mapping of 5Hmu in trypanosoma suggested 5Hmu was enriched in the strand switch, telomeric, and intergenic regions in the nucleosomal genome [58], consistent with 5Hmu inherent-DNA twistability affecting nucleosomal occupancy and may retrospectively preserved the non-nucleosomal state during LCC evolvement.

LCCs had a high frequency of methylcytosine (~1%) [52], which will foster Z-DNA in the presence of high counterions [56,59,60]. Z-DNA, which occurred at a high frequency in dinoflagellate genomes [61] are generally associated with double-helix buckling (commonly referred to as “kinks”) at junctions between supercoiled domains, which could be the result of protein-binding or transcription, effectually dissipating superhelical torsion between domains [62,63]. This would likely, in conjunction with inter-domain cations, contribute to torsional sustainability between SPMs.

## 5. Conclusions and Future Perspectives

Up to 200 metres of DNA can be packaged into a dinokaryon; LCCs embody superhelical-liquid crystalline condensation for compacting some of the largest eukaryotic genomes. This partnership relies on multiphasic phase transitions in the crowded dinokaryon to coordinate multiple levels of chromosome structure. The selective placement of special bases, and high concentrations of selective cations, substantiate higher degrees of DNA superhelicity and anisotropy, likely trending with SPM modularity to reciprocate phase-transitional reversibility and dynamicity. CHLCM embraces modular SPMs, SPM coil-spiraling, special-base(s) placement, high concentrations of counterions, and PGK to optimize soft-matter phase transitions for three special compartments with different chromosomal functions.

## Figures and Tables

**Figure 1 microorganisms-07-00027-f001:**
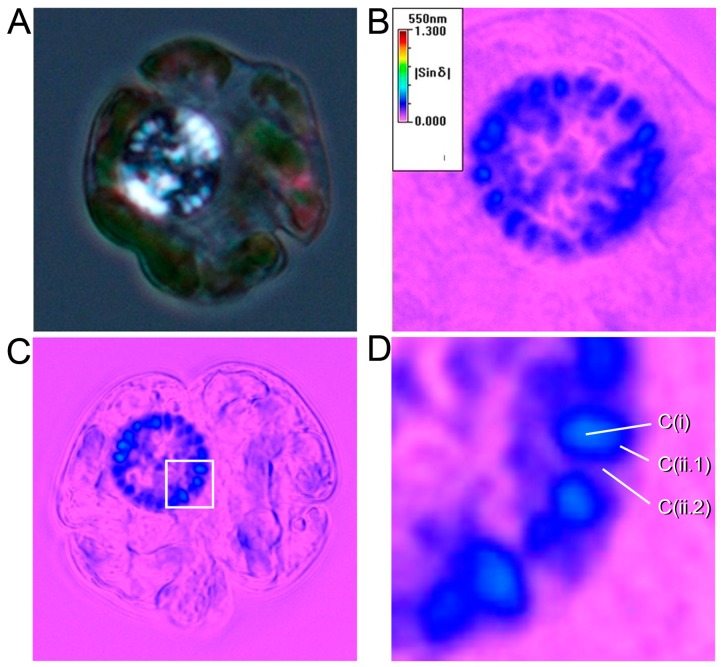
Birefringence imaging of live dinoflagellate cells. (**a**) Birefringence image of a live *Karenia brevis* cell. This cell was in the G_1_ phase of the cell cycle. Some focused LCCs exhibited inner and outer cores. They were observed under cross-polar with Olympus BX-51. A *Karenia brevis* cell has a flattened shape and is approximately 35-50 µm in width. The black and white birefringence within a LCC was indicative of opposite anisotropic orientations in the inner core and outer core compartments (**b,c**) and shows retardance images of a live *Karenia brevis* cell. Retardance, the intensity of the integrated birefringence, was quantitatively imaged through a semi-automatic polarizing light microscopic system, as previously described and displayed in pseudo color [2,11]. The Metripol polarizing light microscope system (Oxford Cryosystem) employed a motorized rotating polarizer and a fixed circular analyzer with Olympus BX-51 [12,13]. Higher retardances were observed at the inner cores (C(i), light blue), when compared to surface compartments (C(ii.1) + PCL C(ii.2), dark blue) in some LCCs. Each LCC in focus was individually surrounded by a third region with low or no detectable retardance, representing C(ii.2) (images by Mike Bennett).

**Figure 2 microorganisms-07-00027-f002:**
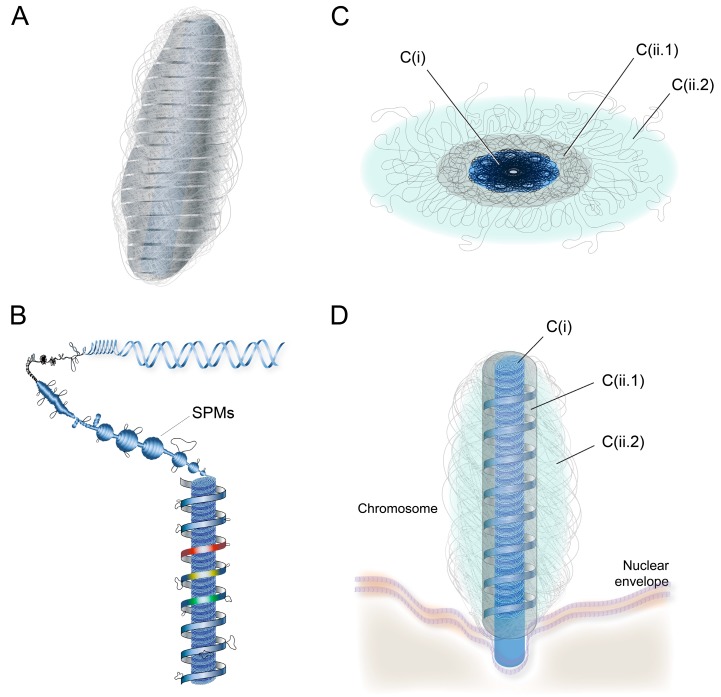
Proposed compartmental superhelical liquid crystalline model for dinoflagellate liquid crystalline chromosomes. Diagrammatic representation of (**a**) a chromonema composed of spiraling supercoiled plectonemic modules (SPMs); (**b**) lateral section of a highly condensed/compacted C(i) superimposed under C(ii.1), composed of chromonema coil-spiraling (partially decompacted to show C(i)); (**c**) the horizontal section of a LCC showing core C(i), the surface C(ii.1) and C(ii.2) consisting of peripheral extrachromosomal loop (PCL); and (**d**) a LCC on nuclear envelope (NE) during dinomitosis; partially decompacted to show C(ii.1) chromonema spiraling, around C(i). (figure drawn by Alvin Kwok).

**Figure 3 microorganisms-07-00027-f003:**
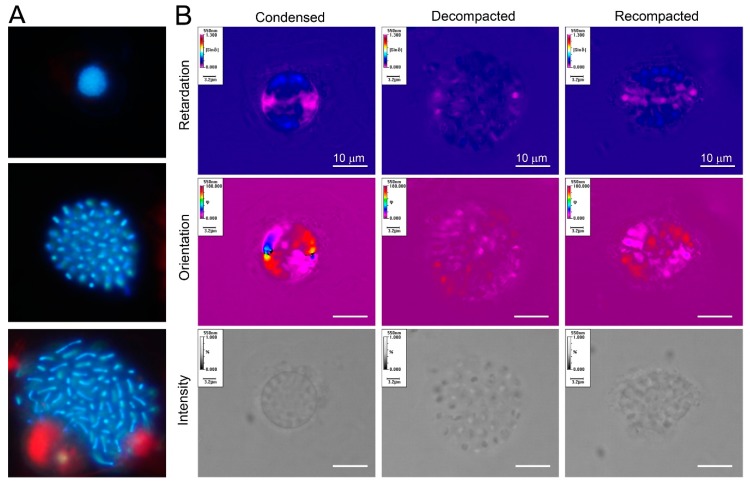
Cation-mediated decompaction and recompaction of dinoflagellate liquid crystalline chromosomes; (**a**) florescence photomicrographs (DAPI-stained) of a compacted and two decompacted nuclei of *Heterocapsa triquietra*. The size of the nucleus is approximately 5 μm. LCCs became extended upon decompaction; (**b**) Metripol images of re-packed and decompacted-LCCs ex vivo. LCCs were decompacted with EDTA treatment and partially re-packed with divalent cation re-addition. The re-packed LCCs, though having regained some compaction, lost their original orientation order. The size of the nucleus was approximately 8–20 μm [32]. (Images by Man Ho Chow and Shiyong Sun).

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
