# Peer review of "Architectural Organization of Dinoflagellate Liquid Crystalline Chromosomes"

_microorganisms, 2019, doi:10.3390/microorganisms7020027_

Round 1
Reviewer 1 Report
This paper explains hypothesized structures of dinoflagellate liquid crystalline chromosomes. The explanations are summarized very well and will be useful for understanding the mysterious structures and the diversification of eukaryotic chromosomes. However, it was difficult to comprehend “SPMs and compartmentation” in the section 2 by the uses of several unclear abbreviations and references. For example, Fig. 2 and Fig. 3 are not referred in the main text.
Specific points:
1. In abstract, C(i) and c(ii) are not defined. “Recent and reinterpretation of previous biophysical data” is correct?
2. In keywords, both “;” and “,” are used.
3. Page 1, line 27: Revise to Fig. 1A.
4. Page 2: Is the model of Fig. 2 hypothesized in this review? Or, show the references in the legend (including Ref. 14). Fig. 2B looks schematic diagram of lateral view, not vertical cross section.
5. Pages 2 and 3: The uses of bold letter in figure legends are unequable.
6. In Fig. 3 legend: “Florescent” may be typo.
7. Page 3, line 13: “nirefringence” maybe typo.
8. Page 3, lines 31-34: Cholesteric CD signal should be explained. CD may be circular dichroism.
9. Page 4, Lines 9 and 10: Can you explain the differences between PCLs and ECLs?
10. Page 4, lines 22 and 23: SIMS should be spelled out.
11. Page 4, lines 28 and 44: What is genome physical karyotype? GPK is a new terminology for me. Can you referred to the proper paper and explain it?
12. Page 4, line 29: “In addition to 5Hmu” is strange in the sentence because 5Hmu is explained in the next section.
13. Page 9, lines 8 and 9: Why 5-hydroxymethyluracil is underlined?
14. Page 5, lines 33-35: I think that the sentence is too much discussion because the ecology and evolution of dinoflagellates are not introduced in this manuscript. You may add to the sentences for the future perspective, not further discussions.
15. Page 6: Refs 11 and 12 are the same papers.
Author Response
Specific points:
In abstract, C(i) and c(ii) are not defined. “Recent and reinterpretation of previous biophysical data” is correct?
amended
2. In keywords, both “;” and “,” are used. amended
3. Page 1, line 27: Revise to Fig. 1A. amended
4. Page 2: Is the model of Fig. 2 hypothesized in this review? Or, show the references in the legend (including Ref. 14). Fig. 2B looks schematic diagram of lateral view, not vertical cross section.
Amended the model now added to both the text and the abstract (highlighted)
Lateral view amended; typos in figure legends corrected
5. Pages 2 and 3: The uses of bold letter in figure legends are unequable. amended
6. In Fig. 3 legend: “Florescent” may be typo. amended
7. Page 3, line 13: “nirefringence” maybe typo. amended
8. Page 3, lines 31-34: Cholesteric CD signal should be explained. CD may be circular dichroism.
Amended (with addition of -265nm)
9. Page 4, Lines 9 and 10: Can you explain the differences between PCLs and ECLs?
Amended to PCLs
10. Page 4, lines 22 and 23: SIMS should be spelled out. Amended to secondary ion mass spectroscopic
11. Page 4, lines 28 and 44: What is genome physical karyotype? GPK is a new terminology for me. Can you referred to the proper paper and explain it? Amended to english “physical karyotype”
12. Page 4, line 29: “In addition to 5Hmu” is strange in the sentence because 5Hmu is explained in the next section. deleted
13. Page 9, lines 8 and 9: Why 5-hydroxymethyluracil is underlined? De-underlined
14. Page 5, lines 33-35: I think that the sentence is too much discussion because the ecology and evolution of dinoflagellates are not introduced in this manuscript. You may add to the sentences for the future perspective, not further discussions. “future perspective” added
15. Page 6: Refs 11 and 12 are the same papers. 12 deleted
Reviewer 2 Report
This is a glowing review of a really good manuscript. So much for modern technology!
Author Response
Thanks for the help